# Efficacy of simultaneous VEGF-A/ANG-2 neutralization in suppressing spontaneous choroidal neovascularization

Richard H Foxton, Sabine Uhles, Sabine Grüner, Franco Revelant & Christoph Ullmer (iD)

See also: **A Wolf & T Langmann** (2019) and
**JT Regula *et al*** (2016)

## Introduction

Currently, anti-VEGF-A therapies are the standard of care for a number of retinal neovascular diseases, including neovascular age-related macular degeneration, and diabetic macular edema (Kim & D'Amore, 2012). However, additional ANG-2 neutralization (Kienast *et al*, 2013; Regula *et al*, 2016) or Tie-2 activation (Campochiaro *et al*, 2016) may be a way to enhance efficacy of treatment for these diseases. We aimed to replicate an *in vivo* study (Regula *et al*, 2016) with a bispecific anti-VEGF-A/ANG-2 antibody in a spontaneous chronic CNV mouse model using JR5558 mice (Nagai *et al*, 2014), whose phenotype has been described differently elsewhere (Hasegawa *et al*, 2014). Our independent study confirms the previously published data. Simultaneous VEGF-A and ANG-2 neutralization was found to reduce neovascular leakage, the number of subretinal inflammatory cells, and retinal cell apoptosis more effectively than either agent alone.

## Results

### Relative efficacy of anti-VEGF-A/ANG-2 combotherapy and anti-VEGF-A monotherapy in a mouse model of ocular neoangiogenesis

Prior to starting dosing of antibodies which were adjusted to equimolar binding sites, JR5558 mice were distributed evenly, in terms of lesion number, across treatment groups. Figure 1B shows no significant differences in pre-treatment lesion numbers between the anti-VEGF-A, anti-ANG-2, and anti-VEGF-A/ANG-2 groups as determined by fluorescein fundus angiography (FFA).

Seven days after the second dosing (Fig 1A), lesion numbers identified during FFA showed strong, significant reductions following treatment with anti-VEGF-A, anti-ANG-2, and anti-VEGF-A/ANG-2 compared with untreated and IgG controls. However, only the anti-VEGF-A/ANG-2 group showed significant differences from both untreated and IgG controls, suggesting an additional benefit with this combination. Nevertheless, there were no significant differences when comparing between anti-VEGF-A, anti-ANG-2, and anti-VEGF-A/ANG-2 treatment groups (Fig 1C).

**Figure 1. Reduction of vessel leakiness and lesion number by combined neutralization of VEGF-A and ANG-2 in the JR5558 mice using FFA.**

(A) Study schematic. Baseline FFA was carried out at P44-45. Antibody injections were given IP on P46 and P53. Post-treatment FFA was at P60 and tissue harvest at P61. (B) Bar/scatter graph of baseline lesion numbers. Lesions were counted in all mice prior to study start. Combined total lesions from left and right eyes were calculated, and then, animals were assigned to treatment groups ensuring no statistically significant differences between groups ($P > 0.05$). (C, D) Bar/scatter graphs showing numbers of spontaneously occurring lesions (C) and area by fluorescence angiography (D) after two weekly doses of antibody (IgG, anti-VEGF-A, anti-ANG-2 at 5 mg/kg IP, and anti-VEGF-A/ANG-2 at 10 mg/kg IP) followed by analysis a week after the last treatment. (E) Representative examples of fluorescence fundus angiograms from the untreated (top left), IgG control (top middle; 10 mg/kg), anti-VEGF-A (bottom left; 5 mg/kg), anti-ANG-2 (bottom middle; 5 mg/kg), and anti-VEGF-A/ANG-2 (bottom right; 10 mg/kg) groups. Data information: SEM is shown as error bars with $n = 9$–10 animals (B) or $n = 19$–20 (C, D) eyes per group and significance indicated by asterisks using ANOVA (B: $P > 0.05$; C: $P < 0.0001$; D: $P < 0.0001$, followed by Newman–Keul's multiple comparison test in C, D). In (C), untreated is significantly different versus IgG control (*$P < 0.05$) and anti-VEGF-A/ANG-2 (***$P < 0.001$). IgG control is significantly different versus anti-VEGF-A (****$P < 0.0001$), anti-ANG-2 (****$P < 0.0001$), and anti-VEGF-A/ANG-2 (****$P < 0.0001$). In (D), untreated is significantly different versus IgG control (*$P < 0.05$), anti-VEGF-A (***$P < 0.001$), anti-ANG-2 (***$P < 0.001$), and anti-VEGF-A/ANG-2 (****$P < 0.0001$). IgG control is significantly different versus anti-VEGF-A (****$P < 0.0001$), anti-ANG-2 (****$P < 0.0001$), and anti-VEGF-A/ANG-2 (****$P < 0.0001$). Anti-VEGF-A/ANG-2 is significantly different versus anti-VEGF-A (*$P < 0.05$) and anti-ANG-2 (**$P < 0.01$). FFA, fluorescein fundus angiography; P, post-natal day; AB, antibody; IP, intraperitoneal; SEM, standard error of the mean; ANOVA, analysis of variance.

Source data are available online for this figure.

Ophthalmology Discovery, Pharma Research& Early Development, Roche Innovation Center Basel, F. Hoffmann-La Roche Ltd., Basel, Switzerland.
E-mail: christoph.ullmer@roche.com
**DOI** 10.15252/emmm.201810204 | Published online 30 April 2019

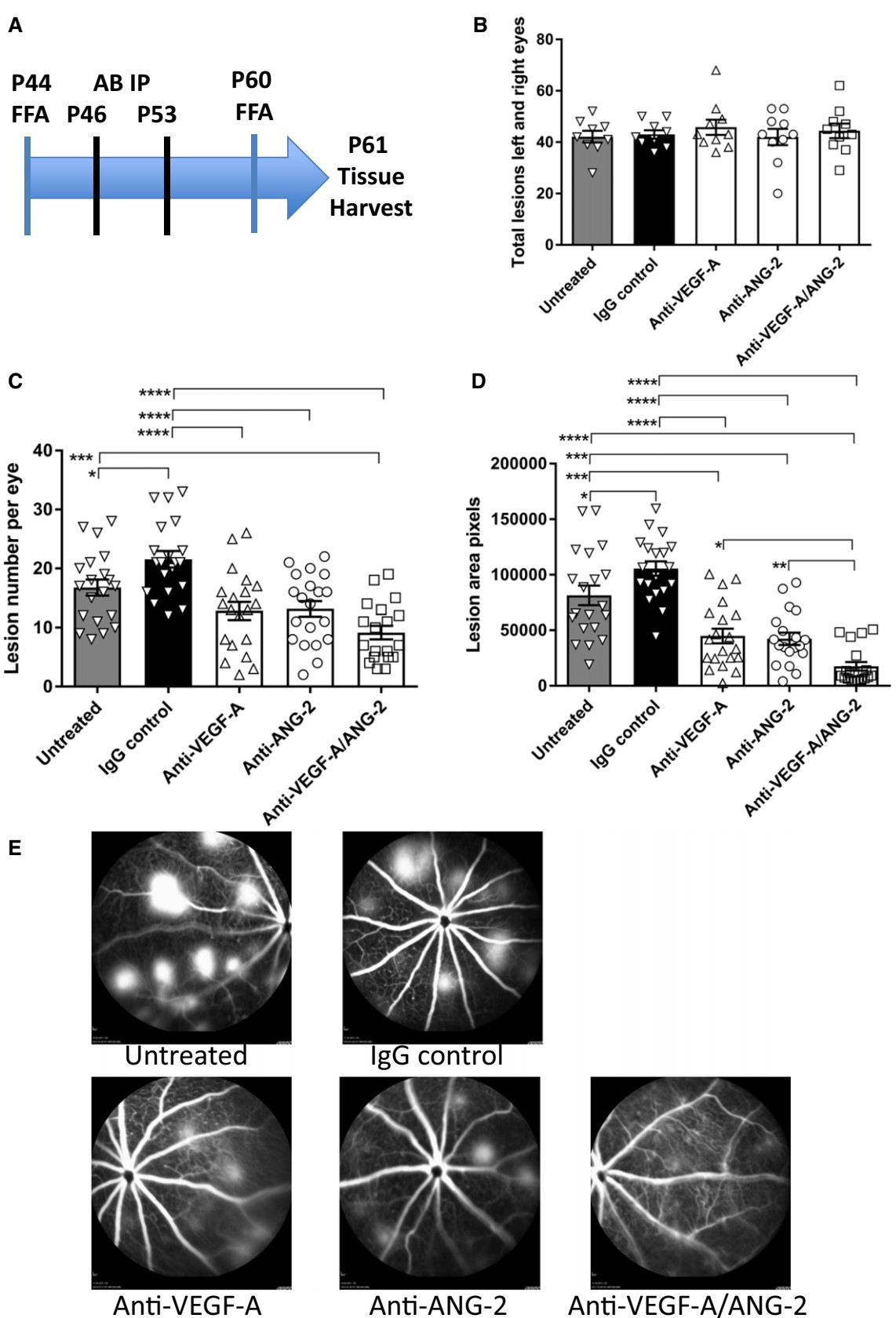

**Figure 1.**

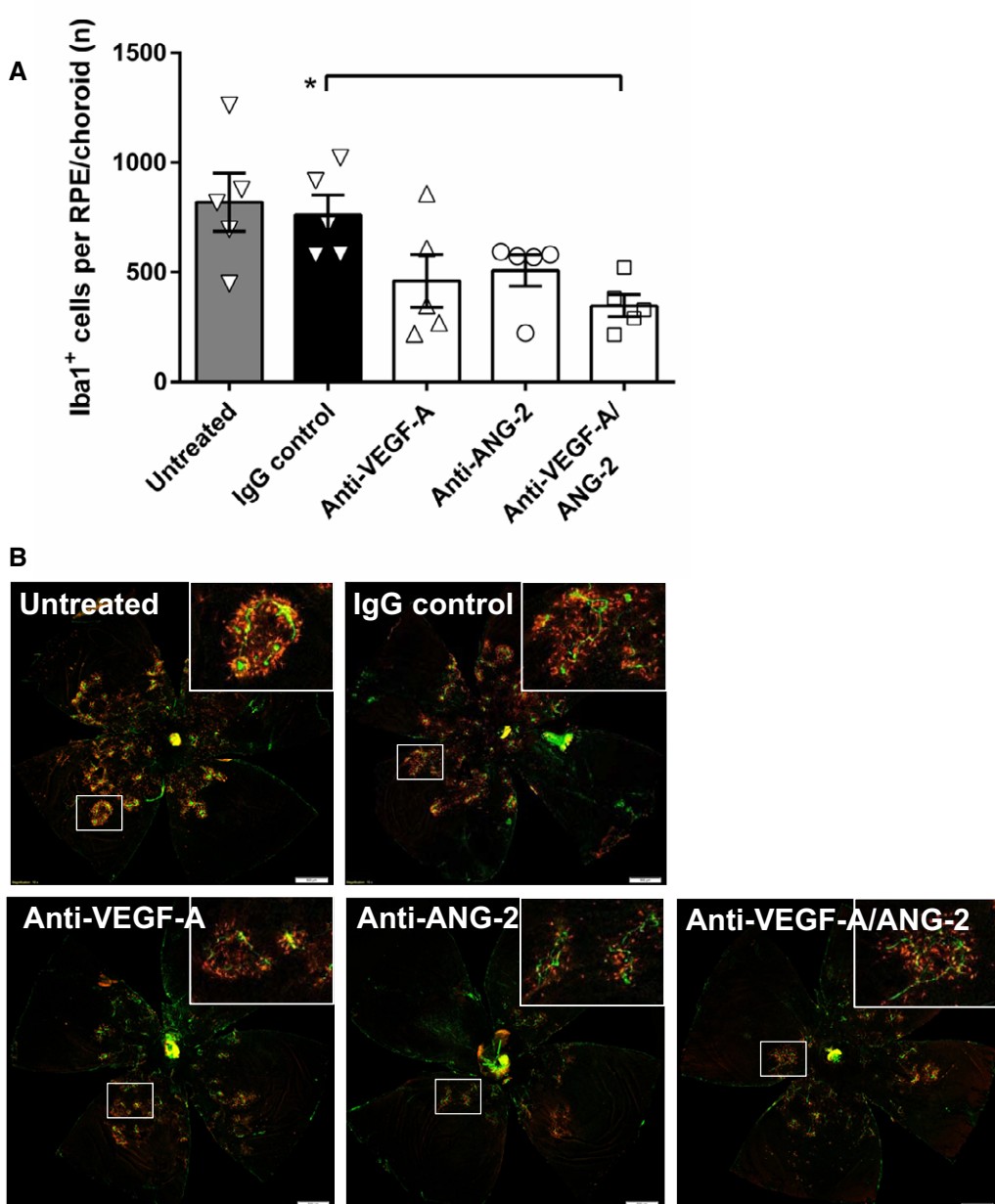

**Figure 2.  Combined neutralization of VEGF-A and ANG-2 reduced the number of Iba1⁺ macrophages in the JR5558 mice.**

(A) Bar/scatter graph of the number of Iba1$^+$ cells in the subretinal space of mice treated with 10 mg/kg IgG control, 5 mg/kg anti-VEGF-A, 5 mg/kg anti-ANG-2, 10 mg/kg anti-VEGF-A/ANG-2, or left untreated. Data are shown as mean ± SEM with *n* = 5 animals per group, and asterisk denotes significant changes after one-way ANOVA and Tukey's multiple *t*-test (Tukey–Kramer HSD). Anti-VEGF-A/ANG-2 is significantly different from IgG control (*$P$ < 0.02). (B) Representative examples of flat-mounted RPE/choroid Iba1 (red) and IB4 counterstain (green) of untreated (top left), IgG control (top middle; 10 mg/kg), anti-VEGF-A (bottom left; 5 mg/kg), anti-ANG-2 (bottom middle; 5 mg/kg), and anti-VEGF-A/ANG-2 (bottom right; 10 mg/kg). Adjustment of brightness to 100 using Photoshop CS6 was applied equally to all images, and non-tissue background was removed using Lasso tool. Inserts show selected areas. Scale bars = 500 μm. SEM, standard error of the mean; ANOVA, analysis of variance.

Source data are available online for this figure.

When lesion areas were analyzed, significant reductions were observed when comparing anti-VEGF-A, anti-ANG-2, and anti-VEGF-A/ANG-2 to both untreated and IgG controls. Furthermore, significant decreases were also observed in the anti-VEGF-A/ANG-2 group compared with anti-VEGF-A and anti-ANG-2 monotherapies, demonstrating an additive effect of the combination treatment (Fig 1D and E).

**Combined neutralization of VEGF-A and ANG-2 reduced leukocyte infiltration**

Administration of bispecific anti-VEGF-A/ANG-2 significantly reduced the amount of

Iba1-positive macrophages on flat-mounted RPE/choroid and around the lesions by 54.4% compared to IgG control, while anti-VEGF-A (39.6%) and anti-ANG-2 (33.4%) showed a strong but not significant effect (Fig 2A and B; Appendix Fig S2).

### Combined neutralization of VEGF-A and ANG-2 decreased retinal neuronal apoptosis

Monotherapies consisting of anti-VEGF-A or anti-ANG-2 led to a mild reduction in TUNEL-positive cells in the outer retina (most likely photoreceptor nuclei), whereas anti-VEGF-A/ANG-2 reduced numbers at a higher significance by 30.4% (anti-ANG-2 by 28.9% and anti-VEGF-A by 18.0%, respectively) compared to IgG control (Appendix Figs S1 and S3).

## Discussion

This study replicated and confirmed previous data showing that simultaneous neutralization of VEGF-A/ANG-2 is superior to either VEGF-A or ANG-2 monotherapies in reducing neovascular leakage, the number of subretinal inflammatory cells, and retinal cell apoptosis in JR5558 mice developing spontaneously developing CNV. This was shown in FFA for neovascular leakage; however, there was no significant effect on lesion number. Number of macrophages as quantified by Iba1 immunostaining was also significantly reduced by the combination of anti-VEGF-A/ANG-2. Apoptosis was most significantly reduced again by anti-VEGF-A/ANG-2 as evaluated by TUNEL. The promising effect of anti-VEGF-A/ANG-2 combination therapy in JR5558 mice adds to the growing evidence that this approach may improve clinical outcomes in patients with retinal diseases beyond anti-VEGF (NCT02484690 and NCT02699450).

## Materials and Methods

### JR5558 mice

Animal experiments were approved by the Federal Food Safety and Veterinary Office of Switzerland (reference BS-2734) and conducted in strict adherence to the Swiss federal ordinance on animal protection and welfare, as well as according to the rules of the Association for Research in Vision and Ophthalmology Statement for the Use of Animals in Ophthalmic and Vision Research guidelines, European Directive 86/609/EEC, and the Roche Ethics Committee on Animal Welfare. Sample sizes for experiments were chosen based on previously published experiments using this mouse strain.

JR5558 mice were supplied from a colony that was also used by Regula et al (2016) and maintained at Charles River, Germany, and sent to F. Hoffmann-La Roche Ltd. at 4–5 weeks old. All animals received food and water ad libitum, in a 12-h day/night cycle, temperature-controlled environment.

At post-natal day (P)44, mice had lesions quantified using FFA to assign to treatment groups, before they received two intraperitoneal (i.p.) doses (1 ml/100 g body weight) on P46 and P53, of either IgG (10 mg/kg), anti-VEGF-A (5 mg/kg), anti-ANG-2 (5 mg/kg), anti-VEGF-A/ANG-2 (10 mg/kg), or were left untreated. B20-4.1 was used as surrogate for anti-VEGF-A (Liang et al, 2006) and LC-10 for anti-ANG-2.

On P60, lesions were assessed again post-treatment with FFA. The following day (P61), animals were sacrificed by $CO_2$ asphyxiation, and eyes were prepared for TUNEL (retina) and Iba1 (RPE/choroid) staining as described below. The study design is shown as a schematic in Fig 1A.

For FFA, JR5558 mice were injected with a subcutaneous mixture of fentanyl (0.05 mg/kg), medetomidine (0.5 mg/kg), and midazolam (5 mg/kg). Eyes were dilated with 0.5% tropicamide (Théa Pharma, Schaffhausen, Switzerland) and fluorescein sodium (2%; Sigma-Aldrich, St Gallen, Switzerland) injected i.p. (13.5 μl/g body weight). After fluorescein injection, animals were placed in front of the Heidelberg Spectralis (HA/OCT Heidelberg Engineering, Heidelberg). Both eyes underwent a complete analysis with a central image and further four images to cover all sections of the eye, which were taken at approximately 7 min (± 30 s) after fluorescein injection. Following imaging, anesthesia was antagonized with naloxone (1.2 mg/kg), atipamezole (2.5 mg/kg), and flumazenil (2.5 mg/kg) and mice recovered. Images were exported manually from the Heidelberg Spectralis Imaging system in JPG format for analysis.

### Tissue harvest

Mice were sacrificed by $CO_2$ asphyxiation at P61 and eyes enucleated and transferred to 48-well plates containing PBS (14190-094; Gibco, Fisher Scientific, France). The anterior segment was removed, and the retina was carefully separated from the posterior segment (RPE/choroid and sclera).

### Iba1 macrophage immunostaining

Enucleated eyes were prefixed with 4% PFA solution for 5 min at RT and transferred to PBS. The RPE/choroid was carefully separated from the retina and transferred to 48-well plates containing 4% PFA and fixed for 1 h at room temperature (RT). RPE/choroid tissue was permeabilized for 2 h at RT in 3% Triton X-100 solution in PBS. After permeabilization, the RPE/choroids were incubated overnight with biotinylated isolectin B4 (IB4, Griffonia Simplicifolia, 1:100, #L2140, Sigma-Aldrich, St Gallen, Switzerland) and goat anti-Iba1 antibody (1:100, #Ab5076, Abcam, Cambridge, UK) in 0.3% Triton X-100 with 5% donkey serum in PBS. The following day, the tissue was washed 6 × 5 min with 0.3% Triton X-100 in PBS, before it was incubated for 2 h at RT with 1:100 FITC-conjugated DyLight 488 Streptavidin (1:100, #SA-5488, Vector Laboratories, Peterborough, UK) and Donkey anti-Goat IgG (1:200, Alexa Fluor 550, #A21432, Life Technologies, Switzerland) secondary antibody in 0.3% Triton X-100 with 5% donkey serum in PBS. Afterward, RPE/choroid tissue was washed 5 × 5 min in 0.3% Triton X-100 in PBS at RT, before being flat mounted on Superfrost glass slides using Dako fluorescent mounting medium (#S3032, Dako, Cambridge, UK).

### TUNEL staining

Dissected retinas were transferred to 48-well plates containing 4% PFA solution and fixed for 1 h at room temperature (RT). Following fixation, retinas were washed 3 × 5 min in PBS, then permeabilized in 3% Triton X-100 in PBS for 2 h at RT, and washed again 2 × 5 min in 0.3% Triton X-100 in PBS. TUNEL protocol was performed according to the manufacturer's instructions (Promega, Southampton, UK). Retinas were incubated for 15 min in 2 × SSC buffer and then washed 3 × in 0.3% Triton X-100 in PBS. For flat mounting, retinas were incised four

times and reversed flat mounted with photoreceptor layer up on Superfrost glass slides (Thermo Fisher Scientific, Switzerland).

### Data acquisition

*Fluorescence angiography image analysis*
For baseline analysis (from P44-45), the number of lesions in each animal per eye was counted to assign mice to treatment groups. The number of lesions per eye was counted using Adobe Photoshop (version CC 2014; Adobe Systems, CA, USA). Lesion areas were not assessed for baseline. Following quantification, the animals were assigned to treatment groups that had statistically identical numbers of lesions. Animals with low numbers of lesions ($< 10$) were excluded and did not undergo treatment or any post-treatment analyses. To quantify post-treatment lesion number and area, jpeg files from each individual eye were labeled and counted with the Photoshop Count tool. Lesion areas were quantified using the Photoshop Lasso tool and expressed as raw pixel numbers for statistical analysis.

*Semi-automated quantitation of Iba1-positive cells*
To quantify Iba1-positive cells on flat-mounted RPE/choroid tissue, whole tissue images (~5 mm × 5 mm) were acquired with an Olympus VS-ASW scanner equipped with a XM10 camera (Olympus Soft Imaging Solution software) and a UPLSAPO 10×/0.40 objective at 565 nm (Iba1) and 518 nm (IB4). On average, 168 snapshot images per RPE/choroid were stitched to one whole tissue image using the VS-Desktop 2.6 software. Iba1-positive cells were counted using the single point selection tool. Data were exported to GraphPad Prism version 6 (graph and scatter plots).

*Semi-automated quantitation of TUNEL-positive retinal cells*
To quantify TUNEL-positive neurons, tile scans of the whole retina were taken, and the total number of TUNEL-positive cells in the ONL was counted. Images of flat-mounted retinas were acquired using a Zeiss Confocal LSM710 using a C-Apochromat 10x/0.45 W M27 objective at 488 nm. Between 6 and 8 confocal image frames in Z-direction (512 × 512 pixel per frame, distance between frames was 8 μm)

spanned the outer nuclear layer with 40–60 μm. 8 × 8 tiled Z-stacked images were stitched with 10% overlap to create one single retinal layer overview with ZEN2.1 software. A maximum projection of these overview scans was created as CZI format, converted into TIFF, and processed using Fiji ImageJ program. Briefly, the retinal area was selected, region of interest (ROI) measured, and images processed with plugin "Find Maxima" with noise tolerance set ~26–30 for point selection. Results including total area, noise tolerance total counts, and coordinates of selected points were copied to GraphPad Prism 6 (graph and scatter plots).

### Study blinding

All personnel involved in image acquisition, image analysis, and histology were blinded to treatment group ID until raw data were fully processed. During the study, only drug-administering personnel had information about the identity of the treatments animals received.

### Statistical analysis

Statistics for FFA were performed using GraphPad Prism 6, applying one-sided ANOVA followed by Newman–Keul's post-hoc analysis. Outliers were removed prior to statistical analysis using Chauvenet's criteria, with outliers defined as being $\pm 2 \times$ standard deviations of mean. Data are presented as Mean $\pm$ SEM with $*P < 0.05$, $**P < 0.01$, $***P < 0.001$, and $***P < 0.0001$. Statistics for histology were performed using JMP® 10.0.1 version 2 applying one-sided ANOVA followed by Tukey's multiple t-test (Tukey–Kramer HSD). Data are presented as Mean $\pm$ SEM with $*P < 0.05$, $**P < 0.01$.

**Expanded View** for this article is available online.

### Acknowledgements
The authors wish to thank Nadine Colé for help in preparation of experiments, masking of treatment groups, and antibody injections; Mirjana Lazendic for weaning of mice, masking of treatment groups, and antibody injections; and Mathieu Brecheisen for immunohistochemistry. F. Hoffmann-La Roche Ltd.

### Author contributions
RHF: study design, data analyses and statistics, *in vivo* work, immunostaining, and reporting; SG:

*in vivo* work: fluorescence angiography baseline and final imaging, image export, weaning, experimental documentation, and reporting; SU: immunohistochemistry, data analyses and statistics, and reporting; FR: tissue dissection and processing, immunohistochemistry, imaging, and data analyses; and CU: study design and reporting.

### Conflict of interest
RF, FR, SU, SG, and CU are employees of F. Hoffmann-La Roche Ltd.

### For more information
(i) AVENUE: A proof-of-concept study of RG7716 in participants with choroidal neovascularization (CNV) secondary to age-related macular degeneration NCT02484690.
(ii) BOULEVARD: Phase 2 study of RO6867461 in participants with center-involving diabetic macular edema NCT02699450.

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
