## [Review Process File · EMBO Molecular Medicine]

Efficacy of simultaneous VEGF-A/ANG-2 neutralization in suppressing spontaneous choroidal neovascularization

Richard H Foxton, Sabine Uhles, Sabine Grüner, Franco Revelant, and Christoph Ullmer

Review timeline:

Submission date:	14 December 2018
Editorial Decision:	4 January 2019
Revision received:	18 January 2019
Accepted:	6 February 2019

Editor: Céline Carret

Transaction Report:

1st Editorial Decision

4 January 2019

Thank you for the submission of your manuscript to EMBO Molecular Medicine. We have now received the enclosed report from a referee who refereed the initial Regula et al paper and was aware of the situation. As you will see, this reviewer is globally supportive and I am pleased to inform you that we will be able to accept your manuscript pending the following final amendments:

- Please address referee 1 comments, reword the introduction as suggested, and perform the recommended analysis to improve the study.

Please submit your revised manuscript as soon as possible.

***** Reviewer's comments *****

Referee #1 (Comments on Novelty/Model System for Author):

For my further criticism on the JR5558 model see comments below.

Referee #1 (Remarks for Author):

This paper partially recapitulates previously published (flawed) data. Here, this dataset seems to be solid and enough experiments have been carried out for a proper statistical analysis. The proper single controls for the dual-specific VEGF-A/ANG-2 molecule were also provided. In summary, the paper shows that inhibiting VEGF-A/ANG-2 with a single dual molecule prevents neovascularization and microglia reactivity in JR5558 mice.

A major point of criticism is that JR5558 mice used in the experiments were described here exclusively as spontaneous CNV model and that only one reference related to a single lab (bias!) is cited. However, this model was previously described by two groups. The first report (Nagai, cited here) indicated that the ectopic vascular vessels were choroidal in origin and infiltrated the RPE and intraretinal space. The second group (Hasegawa et al., PLOS One 20014) reported that the model captures early stages of retinal angiomatous proliferation (RAP), with intraretinal vessels diving into the subretinal space but not breaching the RPE as required for a CNV.

I think that the authors should correct this bias here and expand the introduction by including the RAP paper. Consequently, the title term "posterior segment vascularization" could be used instead of "CNV".

The manuscript could be further improved if the available Iba1 staining would not only be used to count the absolute number of cells but also to perform a ramification analysis, e.g. by using the grid-cross method or other tools to describe the morphological phenotype of amoeboid versus ramified cells.

1st Revision - authors' response

18 January 2019

A major point of criticism is that JR5558 mice used in the experiments were described here exclusively as spontaneous CNV model and that only one reference related to a single lab (bias!) is cited. However, this model was previously described by two groups. The first report (Nagai, cited here) indicated that the ectopic vascular vessels were choroidal in origin and infiltrated the RPE and intraretinal space. The second group (Hasegawa et al., PLOS One 2014) reported that the model captures early stages of retinal angiomatous proliferation (RAP), with intraretinal vessels diving into the subretinal space but not breaching the RPE as required for a CNV.

I think that the authors should correct this bias here and expand the introduction by including the RAP paper. Consequently, the title term "posterior segment vascularization" could be used instead of "CNV".

We thank the reviewer for this comment and agree that they bring up a valid point. We are aware of the difference of opinion of the vessel origin. Since the intention of this correspondence was the exact replication of the Regula et al. 2016 data, the JR5558 mice used here were originated from the same colony as used in the studies by Regula et al., 2016 and Nagai et al., 2014, which were described as having CNV. For further clarity, we have mentioned the origin.

Page 3, line 30: JR5558 mice were supplied from a colony that was also used by Regula et al. In addition, we have mentioned the Hasegawa paper in the introduction and that the phenotypes are described differently.

The following change has been made, in order to correct an error in the previous manuscript. Page 3, line 9 and 11: change of 'retinal leakage' to 'neovascular leakage'.

The manuscript could be further improved if the available Iba1 staining would not only be used to count the absolute number of cells but also to perform a ramification analysis, e.g. by using the grid-cross method or other tools to describe the morphological phenotype of amoeboid versus ramified cells.

We performed confocal 20x image acquisitions with maximum projection of 5-7 image frames in z-direction (spanning ~7-10 μm) of the available RPE/choroidal Iba1 staining and conducted a semi-automated microglia morphology analysis applying an in-house generated image analysis software program. We concluded that not every Iba1 positive cell could be morphologically characterized. Due to an increased Iba1 positive cell density at or close to the lesions of untreated and IgG controls with numerous clumping cells, it was impossible to identify single cells to assess a ramified/amoeboid shape. Only manually selected single Iba1 positive cells peripheral to the lesions could be analysed which does not comply with an unbiased analysis. Therefore, we expected no clear conclusion from such an analysis since only selected Iba1 positive cells would be analysed per lesion and the number of Iba1 positive cells varied among the different treatment groups, being lowest in the VEGF-A/ANG-2 group.

Corresponding Author Name: Christoph Ullmer
Journal Submitted to: EMBO Molecular Medicine
Manuscript Number: EMM-2018-10204